# Clinical Evidence of Mesenchymal Stromal Cells for Cerebral Palsy: Scoping Review with Meta-Analysis of Efficacy in Gross Motor Outcomes

**DOI:** 10.3390/cells14100700

**Published:** 2025-05-12

**Authors:** Madison C. B. Paton, Alexandra R. Griffin, Remy Blatch-Williams, Annabel Webb, Frances Verter, Pedro S. Couto, Alexey Bersenev, Russell C. Dale, Himanshu Popat, Iona Novak, Megan Finch-Edmondson

**Affiliations:** 1Cerebral Palsy Alliance Research Institute, Speciality of Child and Adolescent Health, Sydney Medical School, Faculty of Medicine and Health, The University of Sydney, Sydney, NSW 2006, Australia; madison.paton@cerebralpalsy.org.au (M.C.B.P.); alex.griffin@cerebralpalsy.org.au (A.R.G.); remy.blatchwilliams@cerebralpalsy.org.au (R.B.-W.); annabel.webb@cerebralpalsy.org.au (A.W.); mfinch-edmondson@cerebralpalsy.org.au (M.F.-E.); 2Department of Paediatrics, Monash University, Melbourne, VIC 3800, Australia; 3Parent’s Guide to Cord Blood Foundation, Brookeville, MD 20833, USA; fverter@his.com (F.V.); pedtiago@gmail.com (P.S.C.); 4Department of Biochemical Engineering, University College London, London WC1N 3QS, UK; 5Cell Therapy Laboratories at Yale, New Haven Hospital, Yale University, New Haven, CT 06520, USA; alexey.bersenev@yale.edu; 6Department of Neurology and Neurosurgery, Children’s Hospital, Westmead, NSW 2145, Australia; russell.dale@health.nsw.gov.au; 7Sydney Medical School, Faculty of Medicine and Health, The University of Sydney, Sydney, NSW 2006, Australia; 8Grace Centre for Newborn Care, The Children’s Hospital at Westmead, Sydney, NSW 2006, Australia; himanshu.popat@health.nsw.gov.au; 9The Children’s Hospital at Westmead Clinical School, The University of Sydney, Sydney, NSW 2006, Australia; 10Faculty of Medicine and Health, The University of Sydney, Sydney, NSW 2006, Australia; inovak@cerebralpalsy.org.au

**Keywords:** cerebral palsy, mesenchymal stromal cells, scoping review, meta-analysis, gross motor outcomes, cell therapies

## Abstract

Mesenchymal stromal cells (MSCs) have been under clinical investigation for the treatment of cerebral palsy (CP) for over a decade. However, the field has been limited by study heterogeneity and variable reports of efficacy. We conducted a scoping review of published and registered reports of MSC treatment for CP, with meta-analysis of Gross Motor Function Measure (GMFM) outcomes to summarize research and provide future recommendations. Thirty published reports and 10 registered trials were identified, including 1292 people with CP receiving MSCs. Most received ≥2 doses (72%) of umbilical cord tissue MSCs (75%), intrathecally (40%) or intravenously (38%), and 31% were treated via compassionate/Expanded access. MSC treatment was safe and meta-analyses demonstrated that MSCs conferred significant improvements in GMFM at 3 − (1.05 (0.19–1.92), *p* = 0.02), 6 − (0.97 (0.30–1.64), *p* = 0.005) and 12 months (0.99 (0.30–1.67), *p* = 0.005) post-treatment. Whilst MSCs are safe and improve GMFM outcomes in CP with large effect sizes, study and participant variability continues to confound data interpretation and limits subgroup analyses. With no published Phase 3 trials and high rates of compassionate access, the field would benefit from well-designed trials with unified outcomes. Additionally, data sharing to enable Individual Participant Data Meta-Analysis would support the determination of optimal source, route and dose to progress towards regulatory approval.

## 1. Introduction

Mesenchymal stromal cells (MSCs) are multi-potent, heterogenous, non-hematopoietic stromal cells that are commonly derived from umbilical cord tissue, bone marrow and adipose tissue [1]. MSCs were discovered >30 years ago and since then, our understanding, characterization and use of these cells within medicine has evolved and been refined [2,3]. MSCs are now under investigation as a treatment for a range of neurological conditions including cerebral palsy (CP). CP describes a group of permanent disorders affecting posture and movement, resulting from damage to the developing brain [4]. Available treatments for CP are limited, with a focus on motor training-based rehabilitation to optimize motor potential. MSCs are emerging as a potential intervention with substantial interest across research and the CP community.

MSCs act via anti-inflammatory and immunomodulatory properties and do not require donor-matching due to low immunogenicity, with limited potential for in vivo differentiation [5]. Taken together, these features have enabled widespread use of repeat dose allogeneic MSC products. Research supports that MSCs mainly produce changes in brain outcomes via paracrine signaling [6]. Systemic administration does not lead to MSCs crossing the blood–brain barrier in significant numbers. Despite this, some studies have administered MSCs directly into the brain, although the majority administer MSCs intravenously and intrathecally [7,8]. Across more than two-decades of preclinical research, MSCs for perinatal and adult brain injury have been shown to be potent immunomodulators, secreting anti-inflammatory chemokines, cytokines and growth factors that result in smaller brain infarct sizes, amelioration of damage to oligodendrocyte progenitors, improved myelination, neovascularization, reduced brain cell death and altered brain and systemic inflammation [9,10,11,12,13]. MSCs also directly interact with immune cells to skew macrophage polarization towards anti-inflammatory phenotypes and downregulate immune responses [14]. An important recent discovery was that MSC cell death is required for functional benefit in a range of preclinical models of inflammatory injury [15]. Specifically, when administered intravenously, MSCs are metabolized via first pass in the lungs where alveolar macrophages engulf MSCs and are rapidly cleared. Without cell death and cell clearance, MSCs have impaired efficacy in immunomodulation and paracrine signaling. Immunomodulation is particularly important when considering how to best treat CP. The available evidence supports that there is a pathogenic role of inflammation in CP which should be considered a comorbidity of the condition, with changes in inflammatory status and immune function likely present years after the initial brain injury that may be amenable to treatment [16].

With supportive preclinical efficacy and therapeutic rationale, it has been documented that nearly 50,000 participants over a range of conditions are being, or have been, enrolled across >1000 registered clinical trials using MSC treatment [17]. Encouragingly, a robust safety profile of MSCs is established across a range of conditions [18]. MSC use for CP has been documented in two systematic reviews with meta-analyses [19,20]. These concluded that, from four controlled studies, MSC treatment improves gross motor function (measured via the Gross Motor Function Measure, GMFM) and Comprehensive Function Assessment in children with CP compared to controls. However, Phase 2 trials have suggested inferior benefit compared to other treatments, such as umbilical cord blood [21]. Whilst the evidence of efficacy appears mixed, children continue to be treated in research across various phases of research, as well as via compassionate/Expanded Access Programs, using various methods, dosing, routes and sources of MSCs. Future research directions remain unclear. We conducted a scoping review to better describe the clinical landscape, the number of people treated in clinical studies, and to provide recommendations to direct research efforts. In addition, we also aimed to report on the treatment effect of MSCs for improving gross motor function measured on the GMFM to help provide up-to-date information on efficacy.

## 2. Materials and Methods

We followed the Population, Concept, and Context keywords search method to formulate our scoping review question [22]. The study protocol was published on the Open Science Framework (DOI 10.17605/OSF.IO/GZ9TF) and is reported in accordance with the Preferred Reporting Items for Systematic Reviews and Meta-Analyses extension for Scoping Reviews Guidelines (PRISMA-ScR) [23]. The PRISMA-ScR Checklist can be found in Appendix A.

### 2.1. Search Strategy

The search strategy was informed via previously published systematic reviews [19,20] and preliminary journal searches to optimize term selection. We ran database searches on 20 May 2024 using MEDLINE (1946 to 20 May 2024), Cochrane Central (The Cochrane Library, May 2024) and Embase (1947 to 20 May 2024) via Ovid using the following strategy: (cerebral palsy).tw; (mesenchymal).tw. Searches were de-duplicated. Registered reports/trials were identified using published methodology with searches of 12 international trial registries up to 2023 [24], that included registered compassionate, Expanded or alternate access programs. Registry searches as well as searches of gray literature via Google Scholar were conducted simultaneously and updated on 13 February 2025. A full electronic search strategy can be found in Appendix A.

### 2.2. Study Eligibility Criteria

Eligible studies included all published clinical reports and registered reports/trials (including any available posted results) across any research phase, as well as publicly available information about registered compassionate, Expanded or alternate access programs. Included reports had to administer MSCs (any source) to participants with CP, for the treatment of their CP, but could include a mixed population with no minimum proportion of CP participants. Conference abstracts, posters and full text articles were included in any language where online translation could be utilized.

We excluded MSC secretome products including extracellular vesicles and supernatant, studies with a treatment arm that exclusively administered a mixed cell therapy (like bone marrow alongside MSCs in the same treatment), umbilical cord blood without specific expansion/enrichment of MSCs, and unexpanded/unenriched bone marrow mononuclear cells. We excluded registered trials that had a status of no longer available, suspended or withdrawn with no available posted results. We also excluded registered trials that did not explicitly recruit those with CP even if the keywords provided contained “cerebral palsy”.

### 2.3. Study Selection

De-duplicated database results were exported into Covidence Systematic Review Software (Veritas Health Innovation, Melbourne, Australia, available at: http://www.covidence.org, accessed on 20 May 2024). Additional de-duplication was conducted before titles and abstracts were screened by two independent reviewers (MCBP, MFE). Full texts of studies were then retrieved and independently assessed for eligibility.

### 2.4. Data Extraction, Analysis and Reporting

#### 2.4.1. All Included Reports

Data from trial registries, as well as those from included reports, were extracted by MCBP into a Microsoft Excel spreadsheet developed specifically for this review. Extracted information included study details (year, location, design, author details and registration), participant details (sample size, age, type/topography and severity of CP) and details of the MSC treatment (source, route, dose, how dosing was calculated and treatment regimen/timing).

#### 2.4.2. For Controlled Studies

Meta-analysis was performed on studies that had a control group and reported on outcomes from the GMFM (including all versions such as the -66 and -88), where data were available from two or more studies collecting results at the same timepoint. The GMFM is a gold-standard outcome measure for gross motor function in CP. Review Manager (RevMan, Version 5.4, The Cochrane Collaboration, 2020) was used to conduct quantitative meta-analysis, with outcome data extracted and analyzed for mean, standard deviation (SD) and sample size. When data were reported in other forms (e.g., median with confidence intervals (CI)), the calculator function on RevMan was used to calculate mean and SD. We used a random-effects model with standardized mean differences (SMD) and 95% CI with two-sided *p* values for each outcome.

Subgroup analysis was performed when there were two or more studies collecting the GMFM with similar features at the same timepoint. Subgroup analysis was used to determine potential sources of heterogeneity based on prespecified factors including participant demographics, MSC source, administration route and dose. Heterogeneity between the studies was assessed using the I2 statistic, where I2 > 50% or >75% was interpreted as moderate or substantial heterogeneity, respectively.

Two review authors (MP, AG) independently assessed the risk of bias using the Cochrane criteria (Cochrane Risk of Bias 2) for studies that were included in the meta-analysis. Other studies were not assessed for risk of bias.

### 2.5. Data Reporting

Studies were reported based on the level of evidence (e.g., defined as being published Phase 2, Phase 1 or unpublished registered trials). Compassionate access reports were defined as those that self-reported as providing treatment under regulated Expanded or compassionate access.

Studies were grouped and reported based on report type, publication status and participant information, as well as MSC details. Records were categorized as reported by original authors, however in the instance of registered clinical trials or published reports with missing information, those without a control group were designated Phase 1, and those with a control group were designated Phase 2. In instances where there were multiple reports of the same participants, study authors were contacted to confirm data. Included studies were cross-checked with recent reviews to validate comprehensiveness and accuracy [19,20,25].

## 3. Results

Following database searches, a total of 360 records were identified. After title and abstract screening, 42 full texts were assessed for eligibility. In addition, 29 reports from registries and gray literature were identified. A total of 30 published reports and 10 registered trials were included in this review [5,21,26,27,28,29,30,31,32,33,34,35,36,37,38,39,40,41,42,43,44,45,46,47,48,49,50,51,52,53]. There was duplication in participants reported across two records [30,44]. These two records are presented with collated data received via correspondence from authors to account for all participants. The PRISMA Flow Diagram of the search process is presented in Figure 1.

### 3.1. Study Characteristics

A summary of the 40 included studies is presented in Appendix A. These comprise 16 case series/reports (15 published, 1 published abstract), eight Phase 1 trials (three published, four unpublished, one published abstract), 12 Phase 2 trials (7 published, 4 unpublished, 1 published abstract), two Phase 3 trials (unpublished) and one compassionate access program across two reports (one published, one published abstract, Figure 2).

### 3.2. Participants Characteristics

The total number of participants with CP treated with MSCs in identified reports was *n* = 1292 (Figure 2). Nearly a third of all participants were treated via compassionate access (31%), with 27% treated in Phase 2 trials. Participants treated ranged in age from 6 months to 45 years (Appendix A). Severity, type and topography of CP varied considerably and was often not reported.

### 3.3. MSC Characteristics and Treatment Regimen

Most participants received MSCs from allogeneic origins (either related or unrelated) (Figure 3). Various sources of MSCs were used, with the most frequent being umbilical cord tissue-derived MSCs. Other sources included bone marrow, umbilical cord blood, adipose and dental pulp. A portion of participants were missing MSC source information (21.9%).

MSC dose varied considerably across reports and could not be collated or averaged due to the wide range of dosing selected, repeat dosing and overall differences in total dose (Appendix A). Ten studies did not provide any dosing information. The method for dose calculation varied, but most participants (69%) received a standard dose of cells regardless of body weight. Some participants did not have information on the number of doses received; however, the majority were reported as receiving two or more doses of MSCs rather than a single dose (Figure 3).

Across all studies, 11 routes of MSC administration (including different combinations of routes) were utilized. The most common routes were intrathecal, intravenous or intrathecal and intravenous (Figure 3). Other lesser common routes included intracranial (1.4% of participants), intraventricular (3.9% of participants), a mixture of routes such as intrathecal in combination with intraparenchymal, or a selection of methodologies across participants in the same study (e.g., intravenous or intranasal or intrathecal, Appendix A). Percentages of participants receiving each type of route within the same study were not commonly reported and therefore not able to be presented in detail.

### 3.4. Meta-Analyses of Controlled Studies

Eight controlled studies were considered for meta-analysis of GMFM [21,27,29,35,38,40]. Six studies had adequate outcome data reported and could be collated for meta-analysis, with all other motor-focused efficacy outcomes listed in Appendix A. Timepoints of data collection varied, with at least two studies available for meta-analysis at 1, 3, 6 and 12 months post treatment.

No effect was detected 1-month after treatment on the GMFM (SMD 0.00 (95% CI: −0.65–0.66) *p* = 1.00, Figure 4), with only two studies available for meta-analysis at this earlier timepoint. A statistically significant improvement in GMFM was detected after MSC treatment compared to controls at 3 − (1.05 (0.19–1.92), *p* = 0.02), 6 − (0.97 (0.30–1.64), *p* = 0.005) and 12 months (0.99 (0.30–1.67), *p* = 0.005) post treatment. Using Cohen’s standard, these effect sizes are considered large. However, substantial heterogeneity was detected across all timepoints, with I2 ranging from 64 to 90%, meaning the observed effect varied considerably across studies.

### 3.5. Subgroup Analyses

There was sufficient data to analyze the effect of repeat dosing (one dose versus more than one dose) and route of administration (intravenous versus intrathecal) at the 12-month timepoint, as well as the effect of MSC source (bone marrow versus cord tissue) at 3 months post treatment. There was insufficient data to explore the effect of participant age, type or severity of CP, or dose of MSCs. The test for subgroup differences did not identify significant differences in effect between studies administering one or multiple doses (*p* = 0.38, Figure 5) or studies administering MSCs via intrathecal or intravenous routes (*p* = 0.47) at 12 months post treatment. Similarly, at 3 months post-treatment, no difference was detected in effect size between studies utilizing bone marrow MSCs versus cord tissue MSCs (*p* = 0.33). However, with 0% residual heterogeneity found (Figure 5), repeat dosing, route of administration and MSC source are all plausible sources of heterogeneity.

### 3.6. Risk of Bias Assessment

Figure 6 demonstrates the results from Risk of Bias-2 assessment from the five studies reported and included in the meta-analysis [21,27,35,38,40]. One study was determined to have some risk of bias in the domains of deviations from intended interventions that resulted in an overall score of some concerns [40]. Two studies had overall high risk of bias from selection of the reported results [21], in addition to deviations from the intended interventions [38].

### 3.7. Safety Assessment

Overall, MSC treatment was deemed safe in all studies that reported on it, with no serious adverse events. However, of the 30 published reports, 7 did not contain any information on the capture of safety or side effects.

Studies using intravenous administration of MSCs included reports of fever, diarrhea, vomiting and dyspnea, and were mainly associated with mild, transient infusion reactions. Meanwhile, intrathecal administration of MSCs included safety events of fever, irritability, headache, low back pain, vomiting, nausea and meningism, aligning with increased or decreased cerebrospinal/intracranial fluid pressure. All events were treatable and resolved within days after treatment. One study that assessed donor-specific antibody formation after treatment noted the development of asymptomatic donor-specific anti-HLA class I antibodies in two participants [21].

## 4. Discussion

In this scoping review of MSCs for CP, we present all published and unpublished reports, as well as perform a meta-analysis of controlled trials with subgroup analysis of efficacy for improving gross motor skills on the GMFM. We identified 30 published reports and 10 registered trials, with a total of 1292 people with CP treated with MSCs. A large portion of treated individuals have participated in early-stage research including case series/reports, Phase 1 and 2 studies. However, whilst two Phase 3 trials have been registered and marked as complete [54,55], these have not been published. Phase 3 trials remain an integral part of establishing definitive efficacy evidence to support regulatory approval. Strikingly, there are two published reports of compassionate/Expanded access of MSC treatment for CP that contribute the highest relative proportion of participants treated to date (31%, *n* = 401). This number is likely underrepresented as compassionate access schemes are not routinely published. It is interesting that teams have moved to exploring implementation and alternative access to MSC treatment via compassionate access when there has not been a single MSC candidate or treatment regimen (including dose, route and dose frequency) driven from preclinical to Phase 3 research. Perhaps this is reflective of the lack of commercial interest in MSCs for CP that is required to support costly and resource intense Phase 3 trials (including Phase 3 registration trials) and implementation. Without commercialization opportunities, depending on the country and its regulatory environments, implementation and widespread availability of treatment may be deemed appropriate. For example, the Conditional Early Approval can be granted after supportive Phase 2 research in countries like Japan [56]. These frameworks can be helpful to fast-track approval of promising treatments and deliver them to those who need them. However, the Conditional Early Approval pathways have also been met with some criticism [57], especially when a treatment becomes available in the absence of proven safety and early efficacy signals. In other settings such as the United States, to achieve regulatory approval and reimbursement, the field must focus on streamlining trials to completed Phase 3 trials for treatment approval. In this context, people with CP should not need to rely on compassionate access for treatment access in the long-term and emphasis should be placed on research establishing the comprehensive safety and efficacy profile of MSC treatment for CP.

Promisingly, MSC treatment for CP is safe, and our meta-analysis of GMFM concluded that MSCs confer a large, clinically meaningful improvement in gross motor function at 3, 6 and 12 months after treatment compared to controls [58]. Notably, whilst the GMFM is a gold-standard outcome measure for CP, and widely measured in intervention trials, gross motor is only one outcome that may be improved by MSC treatment. Indeed, there have been calls for exploration of other outcomes, especially those that are considered of high importance to those with lived experience of CP (e.g., quality of life and social participation) [59]. Whilst outside the scope of this scoping review, early evidence from Phase 1 and Phase 2 trials of MSCs for CP do discuss the benefit of treatment for outcomes such as cognition and language [35,49]. Overall, combining these outcomes with those of interest to our community could substantially improve our understanding of the benefits of MSC treatment for CP and also support increasing cost-effectiveness across multiple domains of improvement. Additionally, whilst we do not expect results of MSC treatment to be lost over time, we note that in this scoping review that most reports do not collect outcome measures beyond 12–24 months post-treatment. Long-term follow-up of MSC treatment within CP studies has not been conducted although clinical research has been active for many decades. Whilst this may be entirely appropriate for some patient cohorts, for younger children who are yet to reach their motor potential or for outcomes that are age-dependent, follow-up studies may support confirmation around the sustained or changing benefit of treatment across a range of important outcome measures over time. Nevertheless, the results from this study on GMFM up to 12 months post-treatment remain a timely and important finding, especially as recent publications have questioned the future of MSC research for CP with variable efficacy reported in comparative analyses with other available cell therapies like umbilical cord blood [21,60]. Whilst these recent studies were not powered to detect changes between the two cell therapy arms, it was determined that umbilical cord blood may have a larger effect size than MSC treatment, with earlier detectable changes in GMFM. However, our meta-analysis confirms a large effect size of MSC treatment across timepoints as early as 3 months post treatment (with SMD around 1), and supports the role of MSC treatment as a strategy for improving gross motor outcomes in CP. Not to mention, MSCs may be a favorable therapy especially considering its low immunogenicity and off-the-shelf features that enable safe and frequent repeat dosing, plus feasible high-dosing targets using a manufactured allogeneic product that can be standardized [61]. These MSC characteristics may support their use across a broader treatment window of individuals (potentially across broader age ranges and severity), and this will be important to investigate in more research with well-defined patient populations.

With such variability noted within Phase 2 research, it was critical to explore the effect of repeat dosing, administration route and source of MSCs in subgroup analysis. For the first time in the published literature, there was sufficient data on the GMFM at 12 months to assess the effect of repeat dosing (one dose versus more than one dose) and route of administration (intravenous versus intrathecal), as well as the effect of MSC source (bone marrow versus cord tissue) at 3 months post treatment. Though, due to high heterogeneity and small sample size, no subgroup differences were detected. Our analysis did reveal, however, that repeat dosing, administration route and MSC source reduced residual heterogeneity to 0%, and are therefore plausible contributing factors to heterogeneity. Whilst we did not have enough data to draw any substantial conclusions, it is likely that these MSC treatment characteristics will impact effect size on the GMFM. We are aware of other trials that aim to assess the comparative effectiveness of different cell therapies and administration routes [62]. However, this study has an unknown status and has not been updated since 2018. It also remains critical to better elucidate the effect of repeat dosing and dose per treatment. For instance, one study included in the meta-analysis with the largest treatment effect [38] administered the highest number of doses within the controlled trials (eight doses). The field would benefit from more well-designed, powered trials with the ability to share de-identified data to contribute to a global individual participant data meta-analysis (IPDMA). This would enable addressing research aims such as identifying the best route, dose and dosing threshold, and identification of responder criteria.

Published IPDMA avoids duplication of research, encourages sharing of findings and reduces the burden on finite research resources, supporting knowledge integration [63,64]. We have recently seen the significance of works of this nature in the field, with the world’s first IPDMA of umbilical cord blood for CP being published to help direct who should be receiving treatment, when and at what dose [65]. All MSCs are not equal, and there is no doubt that product variables can impact efficacy, along with participant characteristics such as age, severity, etiology and type of CP. For instance, the IPDMA for umbilical cord blood revealed that treatment is likely to benefit those of a younger age (under 5) with ambulant CP [65]. It is plausible that similar findings may exist within the MSC trial data. It is often hypothesized that as MSCs are highly immunomodulatory, repeat administration at a younger age when inflammation may be higher could confer greater gains [16,21,66]; however, we do not yet have sufficiently powered clinical evidence to draw any definite conclusions.

With the current absence of individual participant data, global data sharing platforms, notable missing data and alternate strategies such as unifying outcome collection via the development of consensus recommendations for reporting standards in future research may be useful. The MSC field has focused over many decades to develop structured methodologies to better characterize products, and establish consistent nomenclature, functional characterization and biobanking standards [2,67,68]. These methods, combined with a minimum dataset within clinical trials treating CP that extends to participant characteristics, outcome selection and reporting, may also support harmonization in published reports and future clinical research. Similar consensus recommendations for data reporting exist within neurology fields undertaking research in neonatal encephalopathy and adult stroke [69,70,71]. This strategy may assist in overcoming missing data that was noted in this scoping review. It also supports movement to unified intervention frameworks based on preclinical and clinical trial results. In future, this could mean that a standardized MSC dose, source, route, participant groups and study outcomes are prioritized, and this could direct later phase clinical trials.

### Limitatons

We acknowledge several limitations of this research. As we wanted to summarize the entire research landscape, we included both published and unpublished reports, which may have introduced bias and error. For example, information shared in trial registries (including those translated into English) may differ from actual study methodology. Inclusion of case reports and compassionate access programs also contribute lower-quality evidence, and lower reliability. However, to overcome this, we integrated the meta-analysis and synthesized findings from controlled studies with assessment of study quality using Cochrane criteria.

Widespread missing data also limited interpretation of some study findings. For example, many reports did not clearly publish the type, topography or severity of CP. We know that MSCs, like all cell therapies, will not be a one-size-fits-all, and publishing transparent cohort characteristics in research is essential. Data heterogeneity also impacted study comparison, interpretation of findings and our ability to conduct meta-analysis with subgroup analysis to determine the effect of dose, route and source of MSCs. For instance, whilst 72% of participants received more than one dose of MSCs, these ranged from 2 to 32 administrations, and it was not possible to sensibly dichotomize the data. Total dose and route used was also highly variable, with one study administering up to 750 million cells over 8 treatments, given across 4 different administration routes. As research progresses, study methods that enable IPDMA as well as strive for alignment in core study outcomes will help support overcoming data heterogeneity and enable study comparison. Similar recommendations have been made for the neonatal cell therapies sector [72]. Investigating the contribution of route, dose and MSC source, as well as identifying the right treatment candidates, remains a high priority to better enable regulatory approval of MSC products [73].

## 5. Conclusions

Whilst MSCs are safe and effective at improving GMFM in CP, an abundance of methods are being utilized that hinder reliability, synthesis of data and interpretation. Research progression from early trials to Phase 3 trials has not been linear, access to MSCs for CP via compassionate access is high and there are no published reports of Phase 3 research. The field would benefit from head-to-head studies as well as robust later stage trials with unified outcome measures. This can enable collaborative IPDMA with knowledge integration that provides recommendations on preferred cell source, route and dose to progress towards regulatory approval of an efficacious MSC therapy for CP.

## Figures and Tables

**Figure 1 cells-14-00700-f001:**
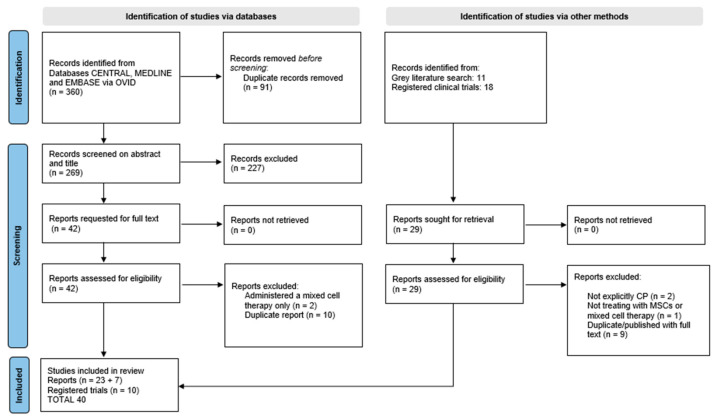
PRISMA Flow Diagram.

**Figure 2 cells-14-00700-f002:**
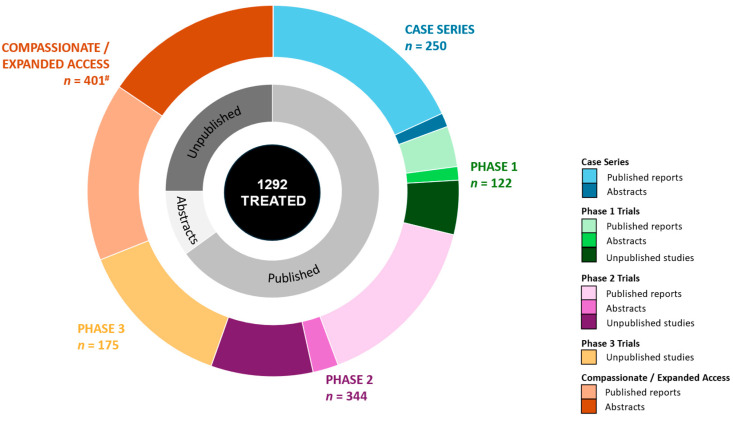
Overview of participants treated by publication status and study type. Total number of participants treated, with breakdown of published/unpublished reports as well as number treated across report type (case series, Phase 1–3 and compassionate/Expanded access). Abbreviations: *n*, number of participants. ^#^ Two reports with compiled sample (*n*) from personal correspondence with authors.

**Figure 3 cells-14-00700-f003:**
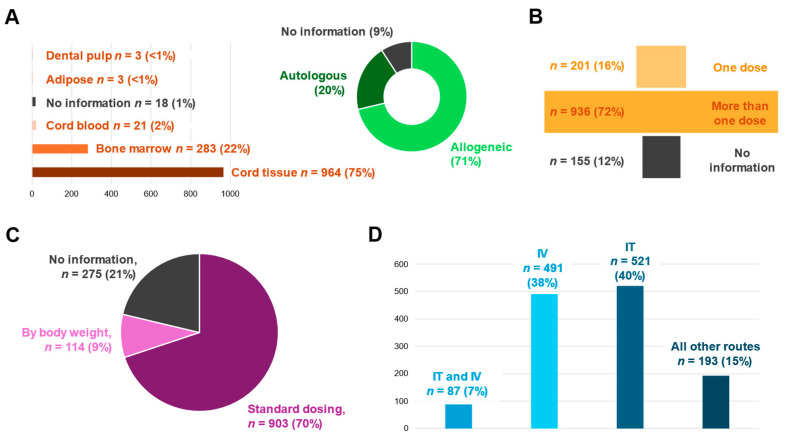
Overview of MSC characteristics and treatment regimens used in included studies. (**A**) Source of MSCs and (**B**) number of treatments with participant numbers reported and proportion of all participants treated (%). (**C**) Report of how dose calculations were determined, either by standard dosing across participants or by participant body weight. (**D**) Total number of participants receiving treatment across the most frequent routes of administration. Abbreviations: IT, intrathecal; IV, intravenous.

**Figure 4 cells-14-00700-f004:**
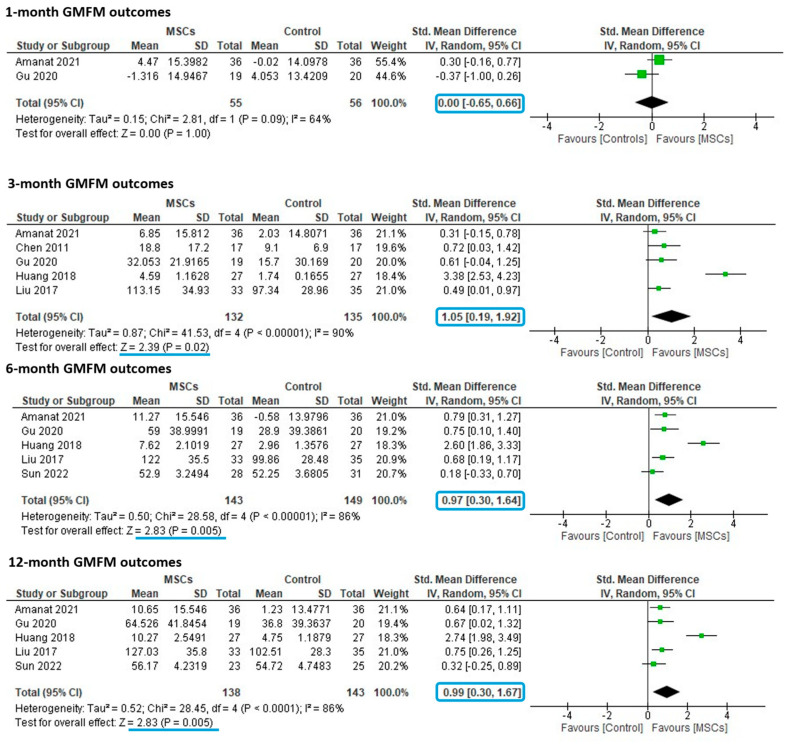
Forest Plots for GMFM at 1-, 3-, 6- and 12- After MSC Treatment [21,27,29,35,38,40].

**Figure 5 cells-14-00700-f005:**
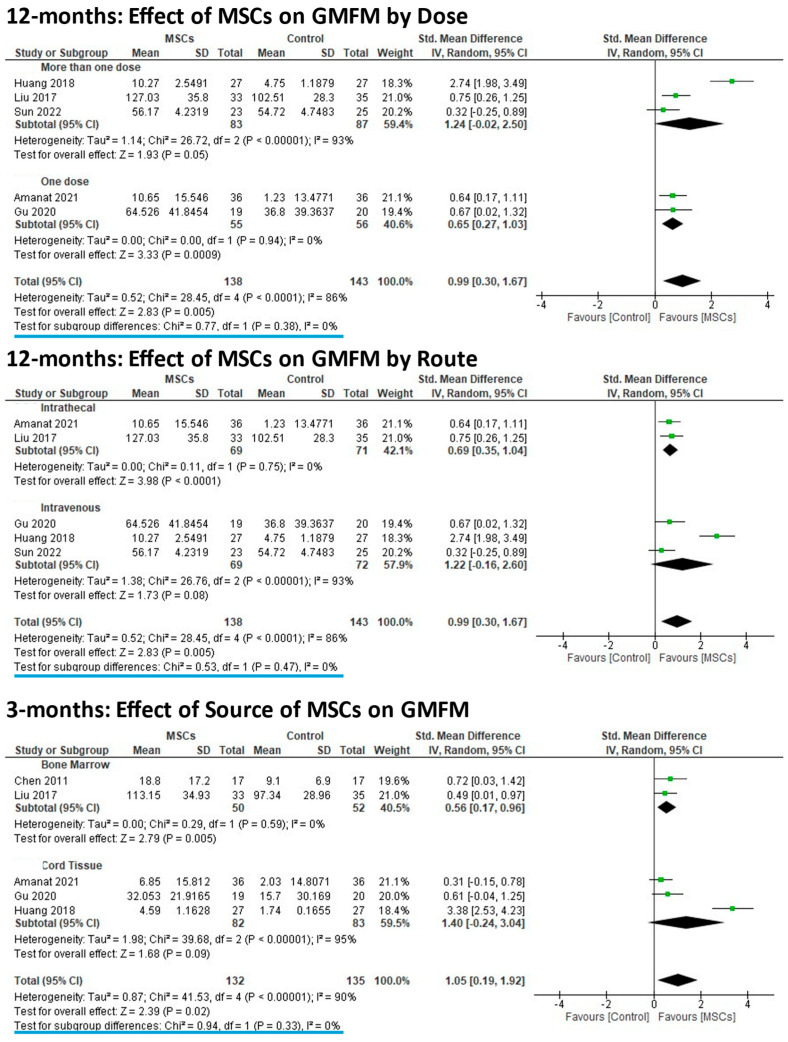
Forest Plots for Subgroup Analyses of GMFM [21,27,29,35,38,40].

**Figure 6 cells-14-00700-f006:**
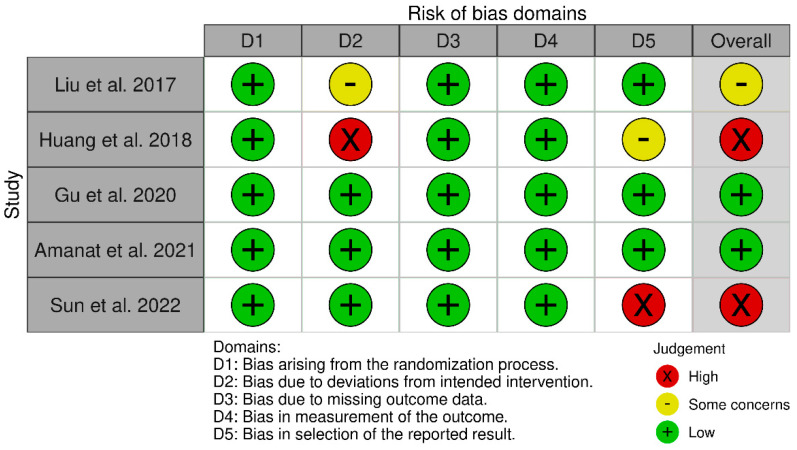
Risk of Bias-2 for controlled studies included in the meta-analysis [21,27,29,35,38,40].

## Data Availability

The original contributions presented in this study are included in the article/Appendix A. Further inquiries can be directed to the corresponding author.

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
