# Peer review of "Clinical Evidence of Mesenchymal Stromal Cells for Cerebral Palsy: Scoping Review with Meta-Analysis of Efficacy in Gross Motor Outcomes"

_cells, 2025, doi:10.3390/cells14100700_

Round 1
Reviewer 1 Report
Comments and Suggestions for Authors
This scoping review with meta-analysis of gross motor outcomes detailed the clinical efficacy of mesenchymal stromal cells (MSCs) for cerebral palsy. Thirty published reports and 10 registered trials were identified through critical review of the available literature which included 1292 people with CP receiving MSCs. The data were solid and the analysis fantastic. Some points are to be clarified before it may become acceptable for publishing.
- The source of MSCs origin were heterogeneous, as shown in Fig.3. Were there different results among the various origins?
- Fig. 4 showed the forest plots for GMFM after MSC treatment. Please describe more detailed results or trends of the outcomes after the treatment. How early the effect could be found after the therapy and the longer the better?
- Would you please provide any changes of inflammatory or pro-inflammatory biomarkers after MSC therapy?
- How about the longterm efficacy (longer than 12 months of follow-up) after MSC therapy since the therapy had been introduced more than 10 years.
Reviewer 2 Report
Comments and Suggestions for Authors
This manuscript is a review and meta-analysis of the clinical application of mesenchymal stem cells (MSCs) in patients with cerebral palsy (CP). This is a scoping review and subsequent meta-analysis, covering 30 published articles and 10 registered clinical trials (total sample size 1,292 people), aiming to evaluate the therapeutic effect of mesenchymal stem cells (MSCs) on gross motor function measure (GMFM index) in children with cerebral palsy. The study pointed out that MSCs treatment had significant therapeutic effects after 3, 6, and 12 months, and had a good overall safety profile. However, limitations such as high heterogeneity, inconsistent study designs, missing data, and no Phase 3 studies being made public weakened the practicality and generalizability of the evidence. However, the manuscript has its value in data integration and analysis.
Comments:
- The clinical trials integrated in this study are highly heterogeneous, including: different sources of stem cells (umbilical cord tissue, bone marrow, fat, etc.); different routes of administration (intravenous, intrathecal, intracranial, etc.); different doses and frequencies (up to 32 doses, and only 1 dose); different ages and disease severity of the subjects, and many studies did not report this information specifically. Even if there is a statistically significant treatment effect, these variables make the results difficult to translate clinically. It is recommended to establish a unified intervention framework and reporting standards (such as standardized cell number/kg for dosage units, age groups, and CP types). The lack of individual patient data (IPD-MA) in current studies limits the ability of stratified analysis. In the absence of a global data sharing system, can alternative strategies be proposed?
- The manuscript points out that currently only the gross motor measurement scale "GMFM" is used as an efficacy indicator. It fails to cover performance indicators such as language, cognition, and quality of life that are more meaningful to CP families. It is recommended that future studies should include "outcome indicators that patients and caregivers consider to be most important" (such as Quality of Life, language development, social participation, etc.).
Round 2
Reviewer 2 Report
Comments and Suggestions for Authors
The authors have made appropriate revisions according to the reviewers' requests.